# Analysis of Pesticide Residues on Fruit Using Swab Spray Ionization Mass Spectrometry

**DOI:** 10.3390/molecules28186611

**Published:** 2023-09-14

**Authors:** Thomas Michael Muggli, Stefan Schürch

**Affiliations:** Department of Chemistry, Biochemistry and Pharmaceutical Sciences, University of Bern, 3012 Bern, Switzerland; thomas.muggli@unibe.ch

**Keywords:** pesticide analysis, quantitation, ambient ionization, swab spray ionization, mass spectrometry

## Abstract

The vast quantity and high variety of pesticides globally used in agriculture entails considerable risks for the environment and requires ensuring the safety of food products. Therefore, powerful analytical tools are needed to acquire qualitative and quantitative data for monitoring pesticide residues. The development of ambient ionization mass spectrometry methods in the past two decades has demonstrated numerous ways to generate ions under atmospheric conditions and simultaneously to reduce the need for extended sample preparation and circumvent chromatographic separation prior to mass analysis. Swab spray ionization enables the generation of ions directly from swabs via the application of high voltage and solvent flow. In this study, swab sampling of fruit surfaces and subsequent ionization directly from the swab in a modified electrospray ion source was employed for the screening and quantitation of pesticide residues. Aspects regarding sample collection, sampling efficacy on different surfaces, and swab background are discussed. The effect of solvent composition on pesticide-sodium adduct formation and the suppression of ionization by the background matrix have been investigated. Furthermore, a novel approach for the quantitation of pesticide residues based on depletion curve areas is presented. It is demonstrated that swab spray ionization is an effective and quick method for spectral library-based identification and the quantitative analysis of polar contact pesticide residues on food.

## 1. Introduction

### 1.1. Ambient Ionization

The generation of ions represents the indispensable initial step of any mass spectrometric experiment, and numerous techniques have been developed for the ionization of a wide variety of compounds of different polarity and mass. While established techniques enable automated analysis protocols and ensure the efficient formation of ions, they usually require some sort of sample preparation. On the other hand, ambient ionization methods circumvent the need for sample preparation as ionization occurs directly on or above the surface of a specimen under ambient atmospheric conditions [1,2,3]. Since the advent of desorption electrospray ionization (DESI) in 2004 [3] and direct analysis in real time (DART) in 2005 [4], a multitude of ambient ionization techniques have been presented [5,6], which have turned mass spectrometry into a versatile and powerful analytical tool for the rapid and hassle-free detection of environmental pollutants, food contaminants, forensic traces, and drug and clinical markers on surfaces [7,8,9,10]. Due to the feasibility of in situ analysis, mass spectrometry has entered novel fields of application, such as homeland security [11] and point-of-care diagnostics [12].

### 1.2. Swab Spray Ionization

Sample collection using swabs and the direct generation of ions from the swabs by applying a high voltage and solvent flow is an attractive approach for pesticide screening, which circumvents laborious sample preparation and cuts down analysis time. Swab spray ionization mass spectrometry was demonstrated for the first time by the Cooks group [13]. This technique, also referred to as swab touch spray mass spectrometry [14], is based on the electrospray ionization (ESI) mechanism and involves the formation of a Taylor cone, a jet region, and a spray plume [15]. Several applications of swab spray ionization have been presented in the past few years, including in the analysis of drugs [13,16], organic gunshot residues [17], and explosives [18,19], as well as in the analysis of biological material such as human gliomas [14], bacteria [20], and newborn screening for nicotine and cotinine [21]. The swab allows easy and effective sample collection on any surface and serves as a depleting analyte reservoir. The applied solvent flow is responsible for continuous analyte extraction during analysis [15]. Due to the rather inhomogeneous texture of swab heads, appropriate choices of solvent, solvent flow rate, applied voltage, and swab positioning are crucial for the formation of a stable electrospray which provides high signal intensity [15]. Swab irregularities and matrix effects can be compensated for by the use of an internal standard, either included in the extraction solvent or otherwise directly deposited onto the swab head [13,14,16,19,21]. Regarding the type of swab head material, rayon swabs featuring an aluminum applicator that conducts the high voltage to the swab head have been demonstrated to be beneficial for many applications [13,14,15,17,18,20,22]. As most ambient ionization techniques are implemented on an experimental level, the modification of commercial electrospray ion sources to meet the specific requirements of swab spray ionization is required.

### 1.3. Pesticide Analysis on Food

According to the Food and Agriculture Organization (FAO) of the United Nations [23], global pesticide usage in agriculture reached 2.7 million tons of active ingredients in 2020. Analytical methods for the detection of pesticide residues on food are indispensable for ensuring food safety at all steps, from the producers to the consumers, therefore avoiding exposure and toxic effects to humans [24,25,26]. Furthermore, the global pesticide usage causes environmental impacts, including the transformation of known pesticides into hitherto unknown agents [27]. 

Most established procedures for quantitative pesticide analysis in food extracts employ GC-MS [28,29,30,31,32] or LC-MS/MS [33,34,35,36,37,38] combined with extensive sample preparation and method optimization to achieve adequate compound separation and to avoid matrix effects. Within the past decade, various ambient ionization mass spectrometric techniques have also been applied to the detection of pesticides in fruit and vegetables. A recent review gives an overview of these applications and discusses the analytical challenges imposed by direct sampling from surfaces, inhomogeneous pesticide distribution, and matrix effects [6]. 

In the current study, swab spray ionization mass spectrometry is evaluated for the qualitative and quantitative analysis of polar contact pesticides on fruit and vegetables. While most ambient ionization methods generate ions directly from a surface or from the atmosphere above a surface, sampling with a swab results in a deposition of compounds, which is continuously depleted during the electrospray ionization process. This intrinsic feature of swab spray ionization requires a novel approach for the quantitation of pesticides based on the extracted ion currents of the depleting analyte ions. Furthermore, sampling efficacy, swab background, analyte extraction from swabs, analyte suppression by the matrix, and sodium adduct formation are discussed. 

## 2. Results and Discussion

### 2.1. Swab Background Evaluation

The head material of commercially available swabs creates a strong background signal, which causes peak overlaps and analyte suppression during analysis. The washing of the swabs prior to sampling and analysis is required. To demonstrate the effect of the washing step on the remaining chemical noise, Copan rayon swabs, as obtained from the manufacturer, were mounted in the ion source. Salicylanilide at a concentration of 1 ppm was fed to the swab head through the capillary. The mass spectra of salicylanilide dissolved in solvent A (Figure 1a) and solvent B (Figure 1b) recorded immediately after application of the spray voltage give evidence for the inherently strong background originating from the swab head material, with ions recorded up to several hundred mass units. The continuous extraction of contaminants from the swab results in a rapid decrease in the background load in the case of solvent A (ethyl acetate/methanol + 0.1% HFo), with only a handful of ions observed after two minutes (Figure 1d). Removing the chemical background also dampens ion suppression, as demonstrated by the extracted ion chromatogram (*m*/*z* 214.07–214.10) of protonated salicylanilide in Figure 1g. Initially, the high background strongly suppressed the formation of this ion, but after spraying for about two minutes the ion current reached a stable plateau at a three-fold-higher abundance level. In comparison, solvent B (methanol + 0.1% HFo) was found to be less effective in eliminating chemical background from the rayon swabs (Figure 1e) and preconditioning for several minutes was required to reduce ion suppression to an acceptable level (Figure 1h). 

These experiments demonstrate the importance of removing background contaminants from the swab heads prior to sampling to avoid overlapping peaks and to decrease analyte suppression. The same cleaning effect as accomplished by the continuous extraction of contaminants from the swab head using electrospray (Figure 1d) was also achieved with a washing step using ethyl acetate and sonication. The efficacy of the washing step is illustrated by the spectra in the right row of Figure 1. Three minutes of washing reduced the swab background to a level comparable to two minutes of electrospray with the ethyl acetate/methanol solvent, as indicated by the full-scan spectra in Figure 1c,f and the extracted ion current trace of protonated salicylanilide in Figure 1i.

To illustrate the advantages in terms of sensitivity and quantitative capabilities, the area of the extracted ion current of salicylanilide was integrated for all three experiments, which yielded 26.1 million counts for solvent B, 181.7 million counts for solvent A, and 467.5 million counts for the prewashed swab in combination with solvent A. Therefore, prewashed swabs in combination with solvent A will deliver the highest sensitivity for quantitative experiments.

### 2.2. Qualitative Workflow

The sampling of the surfaces of an apple and a bell pepper, both purchased at a local supermarket, was performed with prewashed rayon swabs wetted with solvent A. The swabs were wiped over the sample surface for a few seconds and mounted in the holder for direct analysis immediately after sampling. Following the application of the solvent flow and the electrospray voltage, full-scan spectra were recorded for ten seconds and the individual spectra were summed. The identification of compounds was based on the exact masses and the isotopic patterns of the top 1000 ions found in the full-scan mass spectra, and on matching with the high-resolution pesticide precursor ion library. In contrast to LC-MS analysis on reversed-phase stationary phases, ambient ionization is inherently prone to adduct formation, which must be accounted for. The lack of a preceding separation step increases the tendency for adduct formation, and therefore the exact masses of pesticides in protonated form and as sodium and ammonium adducts were considered as well. 

The full-scan mass spectrum shown in Figure 2a gives evidence for the presence of the imidazole fungicide imazalil ([M+H]^+^, *m*/*z* 297.06) and the benzimidazole thiabendazole ([M+H]^+^, *m*/*z* 202.04) on apple skin. The same pesticides were also detected on clementine peel and orange peel, both harvested in South Africa (Appendix A). The carboxamide fungicide boscalid ([M+H]^+^, *m*/*z* 343.03) was found to be highly abundant on the surface of the bell pepper (Figure 2b). Sampling from strawberry leaves gave evidence for fluopyram (Figure 2c), which was detected in protonated form ([M+H]^+^, *m*/*z* 397.05) and as the sodium adduct ([M+H]^+^, *m*/*z* 419.04). Swab contaminants were reduced to a minimum due to the washing step performed prior to sampling. 

Further confirmation of the pesticide’s identity was gathered using tandem mass spectrometric experiments. Product ion spectra obtained through ion trap collision-induced dissociation (CID) were matched against the HRMS pesticide library. Though the lack of upfront chromatographic separation increases the risk of co-isolation of the precursor ions and may lead to contaminant fragment ion peaks in the product ion spectra, the peaks of the most indicative fragment ions dominated the spectra, as demonstrated by the insets in Figure 2. Protonated imazalil ((C_14_H_15_Cl_2_N_2_O)^+^, *m*/*z* 297.06) primarily underwent a loss of the neutral propene moiety (C_3_H_6_, 42.05 Da), resulting in the abundant C_11_H_9_Cl_2_N_2_O^+^ fragment ion with *m*/*z* 255.01. Secondary fragmentation via the opening of the imidazole ring and release of a neutral C_3_H_4_N fragment (54.03 Da) led to the fragment ion detected at *m*/*z* 200.92. If propene is lost in conjunction with the ejection of N-methyl-imidazole (C_4_H_6_N_2_, 82.05 Da), the fragment with *m*/*z* 172.96 will be formed. The release of the dichlorinated tropylium ion (C_7_H_5_Cl_2_)^+^ was responsible for the peak at *m*/*z* 158.98 [39]. Under the applied CID conditions, the fragmentation of protonated boscalid was dominated by the loss of hydrochloric acid from the pyridine ring, resulting in the peak corresponding to C_18_H_12_N_2_OCl^+^ observed at *m*/*z* 307.09. Cleavage of the amide bond released the chlorinated carboxy-pyridine ion C_6_H_3_NOCl^+^ indicated by the peak at *m*/*z* 140.00. The weak fragment ion detected at *m*/*z* 111.58 was generated by release of the chloro-pyridine ion [40]. The product ion spectrum of fluopyram was dominated by the peak at *m*/*z* 208.01, which corresponded to the fragment ion (C_8_H_6_NClF_3_)^+^ obtained by cleavage next to the amide bond. Cleavage of the amide bond yielded the acylium ion C_8_H_4_OF_3_^+^ observed at *m*/*z* 173.02.

### 2.3. Structure and Solvent Effects on the Formation of Sodium Adducts

Structural features of analyte molecules play a crucial role in sodium adduct formation. Sodium ions show a high propensity to bind to the partially negatively charged oxygen atoms of carbonyl groups, e.g., in amides and esters [41]. This is reflected by the experiments performed on apricots harvested in Italy. The spectrum shown in Figure 3a indicates the presence of protonated tebuconazole (*m*/*z* 308.15) and fluopyram (*m*/*z* 397.05), as well as the sodium adduct of fluopyram (*m*/*z* 419.04). Tebuconazole does not contain any carbonyl groups, which explains the very low abundance of the corresponding sodium adduct peak in the spectrum.

However, the extent of sodium adduct formation was found to be affected by the choice of the solvent, as well. The spectrum in Figure 3a was recorded using solvent A, which contains 70% ethyl acetate. Analyte molecules compete with excess ethyl acetate molecules for the available sodium ions, which limits the degree of analyte-sodium adduct formation. In contrast, the spectrum shown in Figure 3b was recorded using methanol as the solvent. Methanol exhibits a lower affinity towards sodium ions and causes fluopyram to be detected almost exclusively in sodiated form. 

### 2.4. Quantitative Analysis

**Depletion curves:** In contrast to flow injection analysis, where the sample concentration remains constant throughout the duration of the analysis, swab spray shows a continuous depletion of sample compounds due to permanent solvent delivery. This requires the analysis time to be taken into account for quantitative work. Quantitative information is obtained from the area under the 7-point Gaussian-smoothed extracted ion current traces (referred to as depletion curves in the following) of the protonated pesticides. The depletion curve of the boscalid [M+H]^+^ ion sampled from bell pepper using a rayon swab and recorded with a continuous solvent feed is depicted in Figure 4a. The signal of boscalid is continuously decreasing due to the extraction of analyte from the swab. Within the first minute of spraying, 40% of the boscalid deposition had already been extracted and the deposition had been reduced to less than five percent after six minutes. The integration of the ion signal over the full analysis time of ten minutes was performed for quantitation. 

**Reproducibility:** The erratic structures of the fibrous swab head material, positional variations of the mounted swabs, and time required for the stabilization of the Taylor cone at the moment of voltage application limited the reproducibility and accuracy of quantitative analysis. Repeated experiments (n = 5) performed by loading 100 ng imazalil dissolved in 4 µL methanol on rayon swabs revealed considerable variations in the shapes of the depletion curves, although recorded under identical experimental conditions (Figure 4b). Complete depletion had already occurred after less than four minutes of electrospraying and the areas under the depletion curves showed deviations within +21% and −14% from the mean area (Appendix A). 

**Calibration curves:** For establishing a calibration curve, prewashed swabs were spiked with 2 µL boscalid reference material at concentrations of 5, 25, 50, 250, 500, 2500, and 5000 ng/µL, resulting in depots of 10, 50, 100, 500, 1000, 5000, and 10,000 ng, respectively. Analyses were performed with solvent A continuously fed to the swab heads at a flow rate of 45 µL/min. The areas under the extracted ion current traces (*m*/*z* 342.90–343.20) of the boscalid proton adduct were considered for establishing the calibration curve (Figure 5, Appendix A). As the integration time intervals were dependent on the depletion kinetics, a 5 min range was chosen for depots up to 1000 ng and a 10 min range for the two larger depots. The LOD was determined using visual evaluation and the Thermo Fisher Scientific Xcalibur genesis algorithm was applied to calculate the RMS signal-to-noise ratio. The limit of detection (LOD) of boscalid was found to be 5 ng (S/N RMS: 82) and the limit of quantitation was 10 ng. The LODs of imazalil and thiabendazole corresponded to 1 ng (S/N RMS: 196) and 2 ng (S/N RMS: 145), respectively (Appendix A).

The calibration curve gave a coefficient of determination of R^2^ = 0.9922 in the range from 10 ng to 10 µg. While the calibration curve is based on defined amounts of reference material depleted from the swabs, the actual amount of sampled analyte depends on its surface concentration, the sampled surface area, and the sampling efficacy. The mass spectrum of the bell pepper sample revealed the presence of boscalid, which was responsible for the peak of high intensity in the spectrum shown in Figure 2b. For fruit skin sampling, an area of 15 cm^2^ (15 cm × 1 cm) was wiped once with a prewetted rayon swab. Data were acquired for 10 min., and the extracted ion current of the protonated fungicide was Gaussian-smoothed and integrated. A comparison of the obtained depletion curve area of 3.47 × 10^8^ counts with the calibration curve resulted in a total of 1386 ng boscalid. The bell pepper weighed 232 g and had a circumference of 24 cm, giving a total surface area of approximately 200 cm^2^. Consequently, the total amount of boscalid fungicide on the bell pepper could be estimated to 82 μg per kg fruit (Appendix A). The European Commission for Health and Food Safety defines the maximum residue level of boscalid on bell peppers as 3 mg/kg [42].

Quantitative results obtained using swab spray analysis may be compromised by incomplete analyte collection from a given surface area and by competitive ionization events leading to analyte suppression. The analyte collection efficacy was found to greatly depend on the surface texture and porosity of the investigated object. Sample recovery was investigated on a glass surface and on fruit skin. Then, 5 µL of an imazalil pesticide standard solution, prepared in methanol, at concentrations of 10 µg/mL, 100 µg/mL, and 1000 µg/mL was applied on 4 cm^2^ glass surface areas, resulting in the deposition of 50 ng, 500 ng, and 5000 ng imazalil. Three repeated samplings (single smears) from the same 4 cm^2^ surface area were analyzed. The resulting individual depletion curve areas were compared to the total signal intensity of all three analyses to represent the sample collection efficacy of a single smear. Recovery from the glass surface was generally high, though it depended on the amount of analyte deposited, with higher surface concentrations giving higher recovery (Table 1). Additionally, the sample collection efficacy on food products including azoxystrobin (sodium adduct, XIC *m*/*z* 425.96–426.26) on tomato skin (Appendix A), harvested in Switzerland, and imazalil (proton adduct, XIC *m*/*z* 297.04–297.07) on orange peel was examined. The first smear on tomato skin delivered 72% of the total signal intensity compared to the next two smears, consisting of only 16% and 12% of the total signal intensity of the three analyses. In contrast to the smooth tomato surface, the sampling of imazalil from the rough and porous orange peel gave a low sample collection efficacy with almost identical portions of 33%, 28%, and 38%. 

These results indicate the variations in sampling efficacy from different surfaces. Swab sampling on rough and porous surfaces is feasible; however, it has a limited ability for quantification only. Regarding the inhomogeneous pesticide distribution on food samples [6], sampling of a larger surface area will be advantageous for obtaining more accurate results.

**Suppression effects:** Due to the lack of upfront chromatographic separation, swab spray ionization is prone to analyte suppression caused by the unavoidable presence of matrix compounds originating from the swab heads or collected from sample surfaces. The extent of analyte suppression depends on the molecular structures, the total amounts, and the depletion characteristics of all compounds present on the swab. Two examples of analyte suppression encountered during swab spray ionization are shown in Figure 6. The XIC chromatograms in Figure 6a correspond to boscalid sampled from bell pepper skin and salicylanilide, which was added at a concentration of 1 ppm to the solvent continuously fed to the swab as a suppression marker for monitoring potentially hindered analyte ionization.

The XIC of boscalid was high at the very beginning of the analysis and declined over the first five to six minutes due to the continuous depletion of material from the swab. No suppression of the boscalid signal was observed. However, the signal of the salicylanilide suppression marker, which was constantly present at a low concentration, suffered from severe suppression and rose from zero to higher abundance over the whole acquisition period of ten minutes. The suppression of salicylanilide may be caused by both the predominant boscalid analyte and the matrix compounds. However, since the decrease in the boscalid signal occurs at a much faster rate than the increase in the suppression marker signal, the matrix is most likely the primary cause of suppression. Because the matrix is depleted as well, the suppression of salicylanilide decreases over time, explaining the curve shape observed. 

The data for the fungicide imazalil were recorded from an orange peel sample, harvested in Spain, and demonstrate the suppression of the analyte by strong matrix constituents (Figure 6b). In addition to imazalil, polyethylene glycol (PEG) was found on orange peel, a compound added by the food industry as a coating agent for extending the post-harvest shelf-life of fruit [43,44]. The PEG signal detected in the range from *m*/*z* 500 to 1000 led to a massive suppression of the ionization of imazalil. The depletion of PEG was found to be rapid and the suppression of the fungicide was significant during the first minute of analysis only. However, the non-simultaneous extraction of analyte and PEG require further investigation to understand the extraction kinetics of analytes of different polarities. 

The presence of sample constituents of relatively high concentration also interferes with quantitative work. The extent of suppression was examined in a series of experiments performed with 10 ng, 50 ng, 100 ng, 500 ng, 1000 ng, 5000 ng, and 10,000 ng boscalid spiked onto swabs and electrosprayed with a continuous solvent flow containing 1000 ng/mL (1.264 ppm) salicylanilide as the suppression marker. This setup resulted in the injection of 45 ng salicylanilide per minute at a continuous flow rate of 45 µL/min. No suppression of salicylanilide was observed for swabs loaded with up to 1000 ng boscalid. An analysis of the samples with higher boscalid load, however, revealed the suppression of salicylanilide within the first two minutes, until the boscalid level decreased to lower values (Appendix A). Nevertheless, these experiments demonstrate two potential causes of analyte suppression during swab spray analysis: the presence of a second abundant analyte and the unavoidable presence of the matrix.

## 3. Material and Methods

### 3.1. Chemicals and Materials

Copan 160C minitip rayon swabs with an aluminum applicator contained in a plastic tube (Copan Italy, Brescia, Italy) were used for all experiments. Glass vials for the preparation of stock solutions and silanized inserts for the preparation of the dilution series were purchased from BGB Analytik AG (Böckten, Switzerland). Methanol and ethyl acetate (LiChrosolv hypergrade LC-MS quality), formic acid (Lichropur LC-MS grade), salicylanilide (99%), boscalid, imazalil, and thiabendazole standards (PESTANAL, analytical standards) were purchased from Merck/Sigma-Aldrich (Buchs, Switzerland). Experiments were performed with either a mixture of methanol/ethyl acetate (30/70, *v*/*v*) (solvent A) or with methanol (solvent B), both containing 0.1% formic acid modifier. 

### 3.2. Instrumentation

The instrumentation was based on a modified nanoelectrospray ion source attached to a Thermo Scientific LTQ Orbitrap Velos (Thermo Fisher Scientific, Reinach, Switzerland) high resolution mass spectrometer, shown in Figure 7. The swabs were mounted to point down in a 45° angle and the position of the swab was adjustable in all three dimensions, which allowed the exact positioning of the swab head for the best signal intensity and stability. The solvent supply capillary was adjustable in three dimensions for exact positioning on the swab head. A syringe pump (Harvard apparatus, Holliston, MA, USA) was used for solvent delivery to the swab head through an electrically grounded PEEK capillary. The swab head was positioned at a horizontal distance of 7 mm from the ion sweep cone of the mass spectrometer. This setting also prevented electrical discharges and the pulsation of the Taylor cone, which would lead to signal disruption. The ion source was equipped with a camera for monitoring the swab position and electrospray formation.

The mass spectrometer was operated in the positive ion mode with a spray voltage of +5.5 kV. Mass spectra were recorded in the range of *m*/*z* 50–2000. The mass resolution was set to 30,000 (m/Δm @ *m*/*z* 400). The temperature of the transfer tube was set to 200 °C to prevent thermal degradation of labile analytes. Salicylanilide as a suppression marker was added directly to the solvent at a concentration of 1 ppm to monitor the swab background and to observe analyte suppression during sample acquisition. Tandem mass spectrometric experiments were performed in the LTQ ion trap with a precursor ion selection width of 1 *m*/*z*, helium as the collision gas, a normalized collision energy of 25%, and an ion activation time of 10 ms. The Thermo Scientific Xcalibur 2.2 (Thermo Fisher Scientific, Reinach, Switzerland) software was used for instrument control and data acquisition, and compound identification was assisted by the Sciex high-resolution MS/MS pesticide library comprising 557 compounds (AB Sciex, Baden, Switzerland). 

### 3.3. Swab Spray Handling Procedure

The handle of the swab was cut a distance of a few centimeters from the swab head and mounted on the swab holder. The solvent capillary was aligned with the higher end of the swab head. Solvents were applied to the swab head at a flow rate of 45 µL/min. As soon as the accumulation of solvent appeared on the swab head, which was observed by the top-mounted camera on the ion source housing, the spray voltage was applied to the swab head via its aluminum handle.

To reduce the background signals, a prewashing procedure of the Copan 160C swabs was performed by adding 3.5 mL ethyl acetate to the swab plastic tube and sonicating the swab in the closed tube for three minutes at 40 °C. The swab was removed from the tube and air-dried before usage. 

### 3.4. Evaluation of Quantitation

A stock solution of imazalil reference material was prepared in methanol at a concentration of 1 mg/mL. Then, 25 µL of the stock solution was diluted by a factor of 40 to yield a concentration of 25 µg/mL. Then, 4 µL of the solution was directly deposited on the prewashed swab to examine the depletion of imazalil during an extraction period of 5 min with solvent A. The procedure was carried out five times in total.

Sampling of defined surface areas was performed with the aid of parafilm masks, which were positioned over the sampling areas. Analyte sampling was performed using horizontal and vertical movements of the prewetted swab head. 

## 4. Conclusions

The monitoring of pesticide residues on food is of high importance to ensure consumer safety. Ambient ionization techniques, such as swab spray ionization, offer the advantages of rapid sample collection from any fruit surface and analysis with minimum sample preparation. Pesticides are separated by mass spectrometry without upfront chromatographic separation, even in the case of rather complex samples. The reliable identification of pesticide residues is achieved by assessing the exact masses of the ions, their isotopic patterns, and the product ion spectra. 

Due to the lack of a chromatographic separation step, swab spray ionization is prone to analyte suppression caused by the presence of abundant sample constituents and the unavoidable matrix. Background ions originating from the swab manufacturing process are easily removed in a washing step prior to sampling. The extent of analyte suppression can be monitored by adding a suppression marker at a low concentration to the solvent flow. Though the formation of sodium adducts can be lessened by appropriate selection of the extraction solvent, this has to be taken into consideration for compound identification and quantitative analysis. Shifting the equilibrium from sodium to proton adducts increases the limits of quantitation and facilitates compound identification via MS/MS. Despite swab manufacturing variations, matrix effects, and the surface-dependent sampling efficacy, the method provides quantitative data through the integration of the area under the extracted ion current trace (depletion curve) of a pesticide ion. The reproducibility was found to be on the order of ±21% and the limit of quantitation was in the low ng range, which seems adequate for a rapid, quite quantitative screening method. 

## Figures and Tables

**Figure 1 molecules-28-06611-f001:**
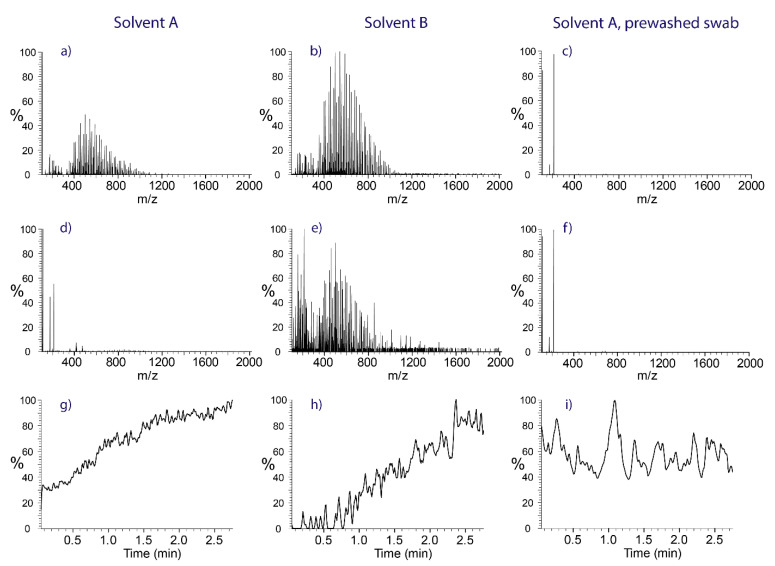
Mass spectra obtained from rayon swabs spiked with salicylanilide, demonstrating the effect of removing contaminations from swabs. **Left row**: Rayon swab as obtained from the manufacturer sprayed with solvent A. **Middle row**: Rayon swab as obtained from the manufacturer sprayed with solvent B. **Right row**: Prewashed rayon swabs sprayed with solvent A. (**a**–**c**) represent the background at the beginning of the measurement, while (**d**–**f**) demonstrate the background at the end of the experiment. (**g**–**i**): extracted ion current traces (gaussian smooth of 7 points) of protonated salicylanilide, indicating the decrease in analyte ion suppression over time.

**Figure 2 molecules-28-06611-f002:**
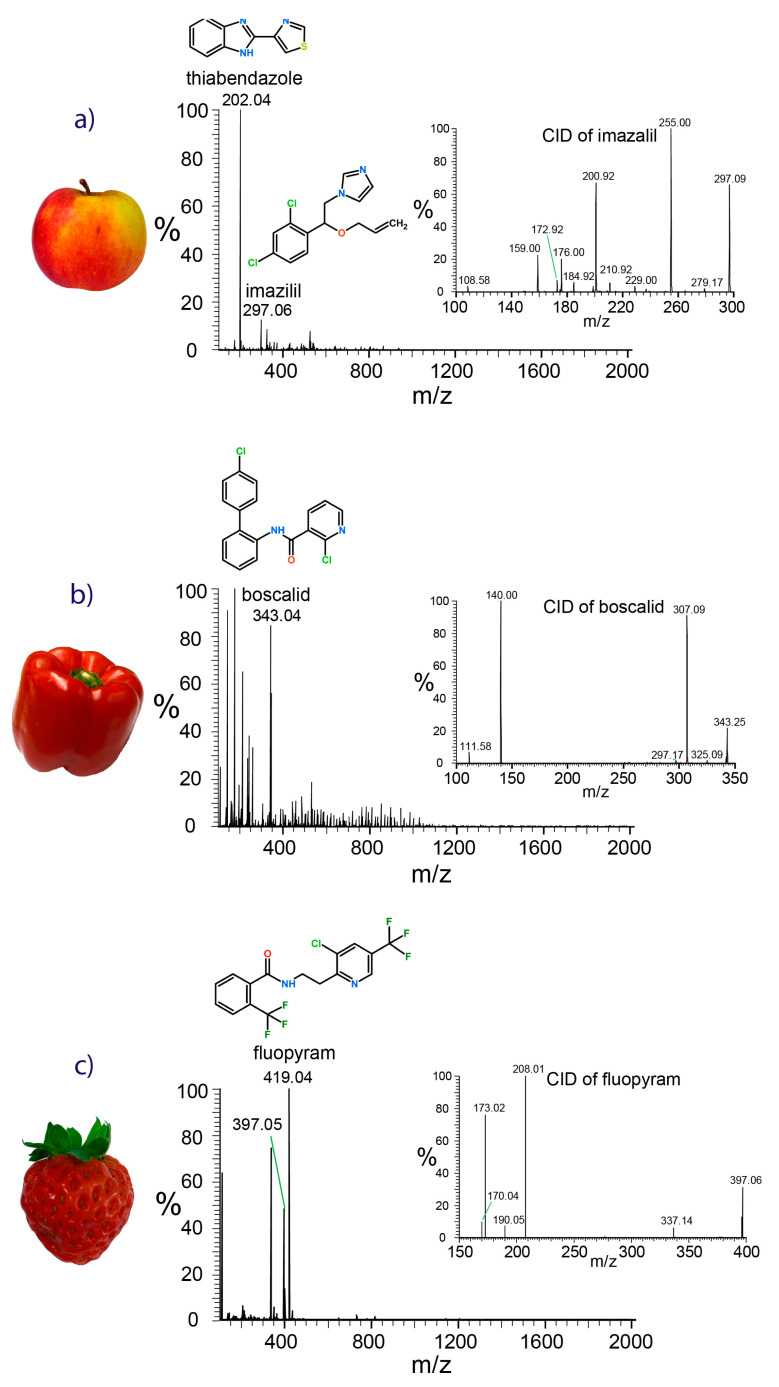
Full-scan swab spray mass spectra of samples taken from the skin of an apple harvested in Switzerland (**a**), a bell pepper from Spain (**b**), and strawberry leaves from Switzerland (**c**), showing the peaks of fungicides. Evidence for compound identities is given by the exact masses, the isotopic patterns, and the product ion spectra (insets).

**Figure 3 molecules-28-06611-f003:**
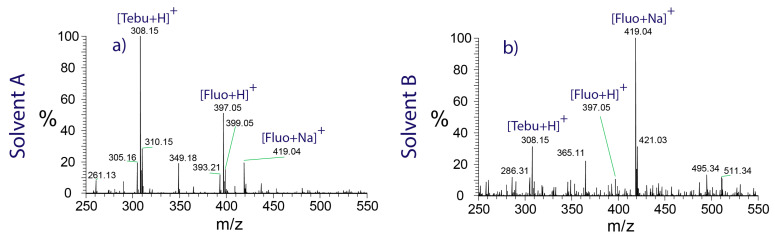
Swab spray ionization mass spectra of an apricot skin sample, demonstrating the influence of the solvent on the formation of sodium adducts. (**a**) Solvent A promoting the formation of protonated analyte, and (**b**) solvent B, shifting the equilibrium to the side of sodium adducts.

**Figure 4 molecules-28-06611-f004:**
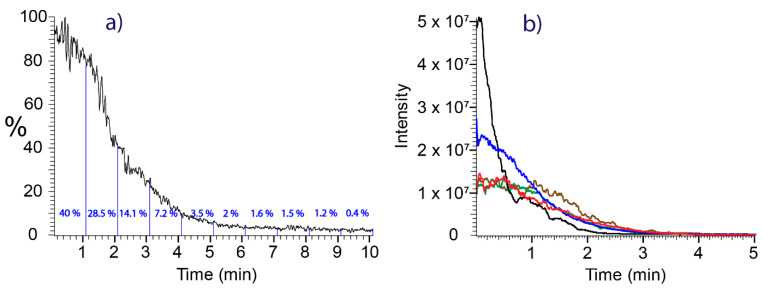
(**a**) Depletion curve (XIC *m*/*z* 342.9–343.2) of the boscalid [M+H]^+^ ion sampled from the skin of a bell pepper. The sections show fractions of material depleted within time slots of 60 s. (**b**) Depletion curves of imazalil reference material (XIC *m*/*z* 297.01–297.10) recorded after spiking 100 ng imazalil dissolved in 4 µL solvent A on swabs. The overlay of five differently colored curves demonstrates the variations in curve shape.

**Figure 5 molecules-28-06611-f005:**
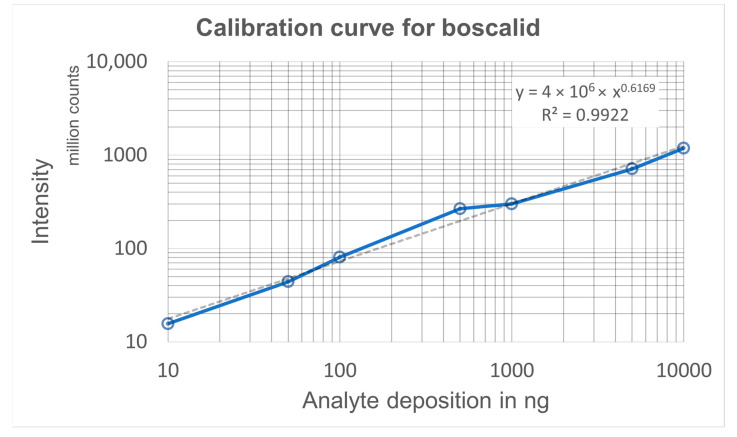
Calibration curve for boscalid.

**Figure 6 molecules-28-06611-f006:**
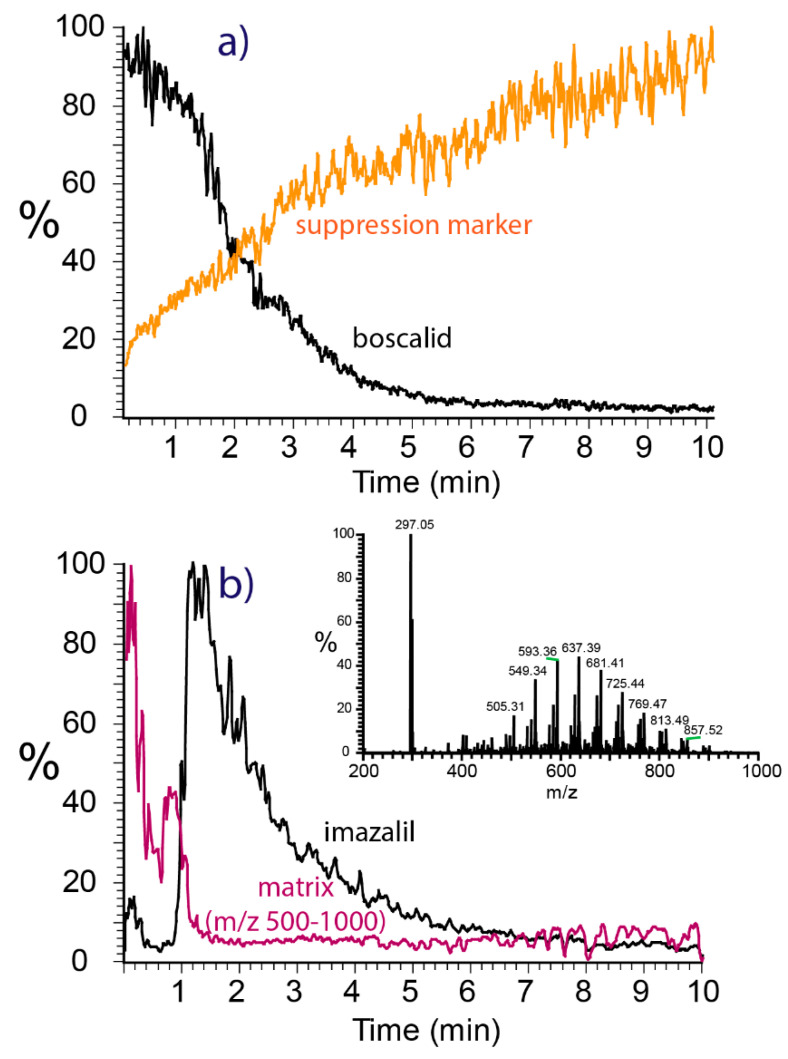
Suppression of ionization encountered during swab spray ionization. (**a**) XIC chromatograms of boscalid sampled from bell pepper and the salicylanilide suppression marker. (**b**) XIC chromatograms of imazalil and the PEG matrix (*m*/*z* 500–1000) sampled from orange peel. The inset shows the full-scan spectrum with the peaks of imazalil at *m*/*z* 297 and the PEG matrix.

**Figure 7 molecules-28-06611-f007:**
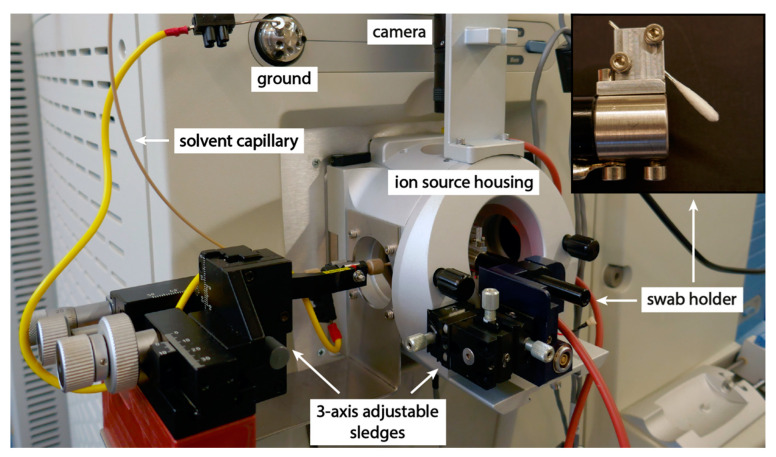
Modified nanoelectrospray ion source with the swab holder and the solvent capillary mounted on 3-axis translation stages for exact positioning. The inset shows the mounting position of the swab.

**Table 1 molecules-28-06611-t001:** Sample recovery rate from different surfaces.

Compound/Surface	Smear (Relative Signal Intensity)		
	1	2	3
Azoxystrobin on tomato skin	72%	16%	12%
Imazalil on orange peel	33%	28%	38%
5000 ng imazalil on glass	76%	12%	11%
500 ng imazalil on glass	68%	20%	12%
50 ng imazalil on glass	41%	38%	21%

## Data Availability

Data related to this work are available from the corresponding author on request.

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
