# Peer review of "Analysis of Pesticide Residues on Fruit Using Swab Spray Ionization Mass Spectrometry"

_molecules, 2023, doi:10.3390/molecules28186611_

Round 1

Reviewer 1 Report

Swab spray ionization is an effective and quick method for the analysis of pesticide residues on food indeed. The manuscript selected two samples for testing. However, the experimental design and summary are subject to be enhanced and do not reach the threshold of acceptability for publication. Here are some advices.

1. The introduction is lack of logic.

2. There are too few research samples. The authors should choose different types of samples, such as rhizomes, leaves, fruits, etc, to demonstrate the general applicability of the method.

3. The experimental design lacked the optimization of conditions for pesticides of different structural types, such as the optimal ionization condition, spray solvent, ionization mode, etc. The experimental design in this manuscript is relatively simple with few samples, and additional experiments is needed.

- The authors should choose different types of samples, such as rhizomes, leaves, fruits, etc, to demonstrate the general applicability of the method.

- The experimental design lacked the optimization of conditions for pesticides of different structural types, such as the optimal ionization condition, spray solvent, ionization mode, etc.

The manuscript has some grammatical errors and needs revision.

Reviewer 2 Report

The entitled article “Analysis of Pesticide Residues on Fruit by Swab Spray 2 Ionization Mass Spectrometry” was carefully reviewed.

 Still it need revision for further consideration.

 Limitations of the work, novelty, and contributions should be highlighted more.

Some more recent relevant articles from Molecules journal can be considered.

 For confirmation of Gas and Mass Spectrometry study the following article needs to be referred.

https://doi.org/10.1002/9781119432241.ch12 

to improve the quality of the paper FTIR spectral data can be considred.

Still there are several syntax errors throughout the paper and these should be corrected. Therefore, the authors are advised to recheck the whole manuscript.

 Overall the article is fairly well written and it may be considered for addressing the above comments.

Need to be improved 

Round 2

Reviewer 1 Report

Accept